# Assessment of ROI Selection for Facial Video-Based rPPG

**DOI:** 10.3390/s21237923

**Published:** 2021-11-27

**Authors:** Dae-Yeol Kim, Kwangkee Lee, Chae-Bong Sohn

**Affiliations:** 1Tvstorm, Sunghyun Building, 255 Hyorung-to, Secho-gu, Seoul 13875, Korea; wagon0004@tvstorm.com; 2Department of Electronics and Communications Engineering, Kwangwoon University, Seoul 01897, Korea

**Keywords:** remote photoplethysmography(rPPG), facial image-based ROI selection, BVP similarity

## Abstract

In general, facial image-based remote photoplethysmography (rPPG) methods use color-based and patch-based region-of-interest (ROI) selection methods to estimate the blood volume pulse (BVP) and beats per minute (BPM). Anatomically, the thickness of the skin is not uniform in all areas of the face, so the same diffuse reflection information cannot be obtained in each area. In recent years, various studies have presented experimental results for their ROIs but did not provide a valid rationale for the proposed regions. In this paper, to see the effect of skin thickness on the accuracy of the rPPG algorithm, we conducted an experiment on 39 anatomically divided facial regions. Experiments were performed with seven algorithms (CHROM, GREEN, ICA, PBV, POS, SSR, and LGI) using the UBFC-rPPG and LGI-PPGI datasets considering 29 selected regions and two adjusted regions out of 39 anatomically classified regions. We proposed a BVP similarity evaluation metric to find a region with high accuracy. We conducted additional experiments on the TOP-5 regions and BOT-5 regions and presented the validity of the proposed ROIs. The TOP-5 regions showed relatively high accuracy compared to the previous algorithm’s ROI, suggesting that the anatomical characteristics of the ROI should be considered when developing a facial image-based rPPG algorithm.

## 1. Introduction

Cardiovascular disease (CVD) is a disease that can affect the heart and the body’s vascular system. Most cardiovascular diseases exist as long-lasting chronic diseases, and there is a lack of appropriate measures to continuously monitor and prevent them [1]. In order to prevent CVD, it is necessary to continuously monitor vital signs for example electrocardiogram, heartbeat, and blood pressure, must be continuously monitored, and professional instruments, such as an IR-UWB heart rate monitor and invasive blood pressure monitor, are required to measure them. However, these devices are for professional use, are expensive, and are not suitable for home use. In addition to professional measuring instruments, there is a method of inferring vital signs, such as heart rate and blood pressure, using an electrocardiogram (ECG). Although electrocardiography is the most accurate method, a photoplethysmography (PPG) method has been developed that can infer the heartbeat in an inexpensive and simple way. PPG is 98% similar to ECG and is an optical technology that requires a single sensor [2]. PPG has become common in recent years and is widely used in wearable vital sign measuring devices, such as smartwatches.

Recently, research on noncontact technology has been progressing beyond contact-type devices, such as wearable devices and heart rate monitors. The photoplethysmography (PPG) measurement method using a facial image is called remote PPG (rPPG) and face PPG (fPPG); rPPG can be measured only with an RGB video camera. Research on rPPG technology was carried out by focusing on the PPG technology of oximeter. PPG is a method of acquiring the pulse waveform of blood vessels noninvasively by using the optical properties of changes in blood vessels on the skin and is used to find out the state of the heartbeat. According to Beer–Lambert’s law [3], the absorbance of a single compound is proportional to its concentration. Hemoglobin has the highest absorbance at the green wavelength, which is a wavelength of 532 nm and utilizes the characteristic that biological tissue reflects and transmits part of the light when the light source is transmitted through the body. The rPPG measurement method using the RGB camera is based on the fact that the extracted value of the ROI from each frame is similar to the PPG waveform [4].

Figure 1 shows a graph of light absorption of deoxyhemoglobin (HHb), oxyhemoglobin (O2Hb), and carbaminohemoglobin (COHb), which are the most abundant in blood. The amount of light absorbed depends on the wavelength of the light, and it shows the greatest absorption at the wavelength of 400-440nm, which is the green channel. The absorption of the wavelength affects the change in the diffuse reflection value, which is responsible for the change in the information received by the RGB camera.

Representative rPPG methods include the ICA [5], GREEN [6,7,8,9], CHROM [10], POS [11], SSR [12], PBV [13], and LGI [14] methods. The ROI selection method is largely divided into a color-based skin detector and a method for designating a chosen area, and there is no clear rationale for this. In this paper, seven representative methods of rPPG are compared with the ROI proposed by each method using pyVHR [15] to provide accuracy. Experiments are conducted using publicly available data, such as LGI [14] and UBFC [16], suggesting that the proposed ROI displays higher accuracy. 

The main contributions of this work are:•Proposal of 31 ROIs that can be used in the rPPG method using an anatomical basis.•Proposal of a BVP similarity (rBS) metric for a performance evaluation in various ROIs.•Performance evaluation of the rBS rank the TOP-5 and BOT-5 using ROI combinations.

The software is available on GitHub (https://github.com/TVS-AI/Pytorch_rppgs (accessed date 26 November 2021)) for experimentation.

This paper is organized as follows. The rPPG methods will be described in Section 2. Section 3 describes the ROI of Section 2’s algorithms and the proposed region of interest. Section 4 presents the experimental results, and Section 5 provides a conclusion.

## 2. rPPG (Remote Photoplethysmography)

The pixels extracted from a face image taken with the RGB camera have face reproduction information, noise, and BVP values. Various methods for extracting the BVP have been studied by analyzing the raw signal in which various information is combined.

Table 1 summarizes representative rPPG methods. As the result of the POS and CHROM method, it has relatively less spread of MAE and PCC values, and highly accurate results can be obtained [15].

## 3. ROI (Region of Interest)

### 3.1. Typical ROI Methods

A facial image-based rPPG algorithm requires a process of finding a face region and selecting an ROI within the found region for efficient signal extraction. Two main methods are used to detect the face area. The most used method is (1) the Viola–Jones method for face detection, which detects a face using the Harr feature [18]. As an alternative to feature-based face detection. there is (2) a skin region detection method [19]. In the past, in the ROI selection process, a method was used based on the face area detected by the Viola–Jones algorithm [20]. This method had the problem of including the background of the border in the ROI in addition to the face area. In another study, using single or additional coordinates within the face area, the forehead, cheeks, and the proposed regions were selected as ROIs [21].

Table 2 shows the ROI selection method of the representative rPPG method mentioned in Section 2. Representative rPPG methods are tried to use the face area as much as possible without focusing on a specific ROI. GREEN and ICA were used for facial image cropping, and CHROM, SSR, POS, PBV, and LGI were used to generate rPPG signals by extracting only specific skin colors.

### 3.2. ROI Analysis Studies

A previous study mentioned that ROI affects signal quality and computational load in the rPPG method [21]. Studies also raised the problem of designating the entire face as an ROI. It was assumed that there would be a protruding part of the blood vessel distribution, and the accuracy was evaluated for the forehead, left and right cheeks, nose, mouth, nasal dorsum, and chin. As a result, the cheeks and forehead were selected as excellent ROIs.

### 3.3. Proposal of ROI Selection

#### 3.3.1. Thickness of Human Face Skin

rPPG is a contrast between specular reflection and diffuse reflection that occurs when light hits the skin. Specular reflections are pure light reflections from the skin, while diffuse reflections are reflections due to absorption and scattering of skin tissue that depend on changes in blood volume [22].

Figure 2 shows the principle of how the camera receives BVP (blood volume pulse) information. When the light source hits the skin, some of the light is absorbed by the skin and blood vessels, and the remaining diffuse reflection information is received by the camera. Depending on the thickness of the skin, the reflection information of the light source can be different. Although blood vessels decrease reflectance and transmittance, diffuse reflection exhibits sensitive dependence on the depth of blood vessels, that is, the thickness of the skin [23]. According to the thickness of the skin, the absorption amount of the light source decreases, which represents a large difference between the specular reflection and diffuse reflection information. The thickness of the dermis and epidermis of 39 anatomical sites of 10 cadavers were measured [24]. The 39 areas used in [24], the relative thickness of the dermis and the epidermis, and the relative thickness of the skin calculated based on the information are as shown in Table 3.

#### 3.3.2. Proposed ROI

To conduct rPPG experiments on the anatomical regions mentioned in Section 3.3.1, we selected the experimental regions. When selecting ROI candidates, the scalp area (temporal scalp, anterior scalp, posterior scalp), ear area (preauricular, upper helix, mid helix, conchal bowl, earlobe, rear ear), and neck area (anterior neck, lateral neck) were excluded. In addition, the area around the eyes (upper medial eyelid, upper lateral eyelid, lower eyelid, and tear trough) was integrated into one region because the size of the region was small. Finally, the symmetrical parts such as the nasolabial fold and marionette fold were divided into two areas, left and right.

Table 4 shows the proposed 31 regions and skin thickness.

### 3.4. Assessment Metric of Proposed ROI

We used three measurement methods used in rPPG to evaluate the performance of the proposed ROIs. In addition, we propose a relative BVP similarity (rBS) method for evaluating the relative superiority of each ROI.

MAE (Mean Absolute Error): MAE was used to see the accuracy of the estimated waveform for each rPPG method.


(1)
$$MAE=\frac{1}{N}\underset{t}{\sum }\left|\hat{S}\left(t\right)-S\left(t\right)\right|\;$$


RMSE (Root Mean Square Error): RMSE was used to view the standard mean error.


(2)
$$RMSE=\frac{1}{N}\sqrt{\underset{t}{\sum }{\left(\hat{S}\left(t\right)-S\left(t\right)\right)}^{2}}$$


PCC (Pearson’s Correlation Coefficient): PCC is a method for interpreting the linear relationship between two given signals. The closer the absolute value of the PCC result to 1, the more linear it is.


(3)
$$PCC=\frac{\sum_{t}\left(\hat{S}\left(t\right)-\hat{\mu }\right)\left(S\left(t\right)-\mu \right)}{\sqrt{\sum_{t}{\left(\hat{S}\left(t\right)-\hat{\mu }\right)}^{2}}\sqrt{\sum_{t}{\left(\hat{S}\left(t\right)-\mu \right)}^{2}}}$$


*S(t)* is the ground truth, and *Ŝ(t)* is the result of the rPPG method. In addition, μ is the average value of S(t), and μ is the average value of *Ŝ(t)*. The results of each of the above three methods were processed to generate the rBS (relative BVP similarity), which is a final evaluation metric.
(4)rBS=(log(max(MAE)−MAE+e)+log(max(RMSE)−RMSE+e))∗|PCC|

In the rPPG method, the MAE is used as a measure to determine the absolute difference value from the actual BVP waveform, and the RMSE is used as a measure to determine the variance value of the difference. The PCC is used to determine the linear relationship between the measured value and the original value. The closer the absolute value of the PCC is to 1, the more linear it is. The waveform of the BVP is significant in extracting ultralow frequency (ULF), very low frequency (VLF), low frequency (LF), and high frequency (HF) well. The included disease information is shown in Table 5.

The smaller the MAE and RMSE values, the more they were shown to be similar to the actual data so that the area with a smaller value is more effective. Because each frequency band means different information, it was designed to have a big impact on the linearity of the waveform.

### 3.5. ROI Assessment Procedure

In order to set the ROIs suggested in Section 3.3.2, three procedures were performed: the Face Mesh Generation, ROI Candidate Setting, and ROI Selection.

Figure 3 is the procedure for assessing the proposed ROIs. The ROI setting was carried out in the preprocessing step of rPPG, and the ROI was created using the landmark created through the face mesh method. Face mesh extraction methods can be divided into cascaded regression-based and deep learning-based methods.

Figure 4a is a face landmark key point of the cascaded regression-based Open Face Project, while Figure 4b shows the face mesh provided by the deep learning-based Media-pipe Project, which are Open-source Face Mesh Projects [25]. In the cascade regression-based method, the representative project open face creates a face mesh with 68 key points and is available in Dlib. As a deep learning-based method, Google’s Media-pipe Project creates a face mesh with 468 key points [26]. In [27], a comparison was conducted with the SAMM dataset composed of various emotion videos of human faces, and the Media-pipe showed high performance with a slight difference. Therefore, in this paper, face landmarks were created using a Media-pipe that can show excellent results in generating various ROIs, and ROI candidates were created by combining landmarks.

Figure 5 shows the result of generating a face mesh image using Media-pipe (a) and the visualization result of the ROI candidate (b).

## 4. Data and Statistical Analysis

The rPPG method is affected by whether the input video is encoded, light uniformity, and skin color. When the video is encoded, the rPPG information is quantized, and the complete information may not be transmitted [28]. If the light is not uniform, the face is not properly detected [29]. The darker the skin color, the lower the amount of diffuse reflection because the melanin content changes [30].

In this paper, the UBFC and LGI-PPGI datasets, which have the least three effects listed above, were selected to verify the validity of the proposed ROIs [14,16]. The UBFC and LGI-PPGI datasets are composed of raw video data and have uniform light brightness.

Figure 6 shows the Fitzpatrick skin color types. Type I means Pale white skin color, Type II means Fair skin color, Type III means Darker white skin color, Type IV means Light brown skin color, Type V means Brown skin color, and Type VI means Dark brown or black skin. In this paper, experiments were conducted with light skin colors of Type I and II among the six skin colors classified on the Fitzpatrick scale. A proposed ROI mask was generated for two datasets, POS and CHROM were applied to the image to which the generated mask was applied, and superiority was verified using the proposed metric.

### 4.1. Benchmark Dataset

UBFC [16]: It consists of 42 videos, heart rate, and a label in which the heart waveform is recorded. The participants looked directly at the camera installed at a distance of 1 m while filming the video and were filmed while solving the given quiz.LGI-PPGI [14]: A video was recorded by giving 6 subjects four conditions: no motion, motion, vigorous motion, and dialogue.

### 4.2. Assessment of Proposed ROI

The results of the experiment with POS, CHROM on the UBFC and LGI-PPGI datasets are as follows. Figure 4 and Figure 5 show the results of performing seven methods on the UBFC and LGI-PPGI datasets by specifying 31 regions. It can be seen that the MAE and RMSE values of region numbers 0, 1, 3, and 27 are excellent regardless of the method type. Figure 6 is the PCC result, and the values of region numbers 0, 10, 27, and 28 show results close to 1.

Figure 7 shows the results of the MAE, RMSE, and PCC metrics on the UBFC data. The yellow boxes show the TOP-5 score, and the blue boxes show the BOT-5 score for each metric. The yellow box indicates the TOP-5 in each metric, it can be seen that region 0 and region 10 are commonly included in the TOP-5 in the whole metrics.

Figure 8 shows the results of the MAE, RMSE, and PCC metrics on LGI-PPGI data. Regions 0, 10, and 27 are commonly included in TOP-5 in the whole metrics. Regions 0 and 10 were found to be the best regions in both datasets.

Figure 9 shows the processed rBS values based on the results of the MAE, RMSE, and PCC. To derive a meaningful BC value, the median value was used, and a meaningful mask was selected as the median value.

Table 6 shows the BS median values for each mask, and as a result, regions 27, 10, 3, 0, and 28 showed high scores, whereas regions 15, 13, 12, 20, and 19 showed low scores in the order. The high-scoring regions have a skin thickness of 1086.2 μm, 1386.11 μm, 1221.88 μm, 1245.63 μm, and 1086.2 μm respectively, while the low scoring regions have relatively thick skin thicknesses of 2015.89 μm, 1794.71 μm, 1794.71 μm, 1496.12 μm, and 1496.12 μm.

Figure 10 is a visualization of the results of Table 5. The yellow areas are the TOP-5 regions, and the blue areas are the BOT-5regions. The white regions are the other remaining 21 regions.

Table 7 is an analysis table for the correlation among the ROIs, the thickness of the skin, and the number of pixels in the region. As a result of Pearson’s correlation, the correlation between skin thickness and rBS rank was 0.50, with moderate positive linearity, and the number of pixels in each region was −0.53, with moderate negative linearity. It can be seen that the thinner the skin and the larger the region, the better the results obtained. According to the results of Table 6, it was shown that there was a correlation with the thickness of the skin and the number of pixels in the region. However, the average number of pixels in the proposed TOP-5 regions is 696 pixels, which is very different from the existing 25,000 pixels used for the entire face. The smaller the region, the easier it is to be exposed to noise, such as light distortion or movement. To solve this problem, a combination of regions was proposed, and an experiment was conducted.

Table 8 shows the region combination of the proposed region and the evaluation results of the existing ROI method. The average thickness of TOP-5 is 1191.11, and the number of pixels is 2431. BOT-5 has an average thickness of 1581.39 and an immersive pixel count of 1030. As a result, the region combination had a positive effect on the improvement of the results, and the proposed TOP-5 combination showed higher accuracy than the Face + Skin method, and BOT-5 showed lower accuracy.

Figure 11 is the BVP extracted from the proposed ROI using the POS method. Yellow is the BVP extracted from the TOP-5 ROI. Comparing it with the blue BOT-5 BVP, the yellow waveform is more similar to the green ground truth. In particular, there is less variability and less noise than the blue waveform.

## 5. Conclusions

In summary, in this paper we have proposed:Proposal of ROI candidates among 31 facial regions through skin thickness and anatomical analysis.A metric called rBS that can be used to assess the excellence of each ROI.

In conclusion, the ROI selection in the rPPG method is as important as the signal extraction method. As rPPG uses diffuse reflection information, it has been demonstrated that the thickness of the skin affects the result. To extract the validity of skin thickness-based ROI selection, 31 masks and rBS metrics were proposed. For the UBFC and LGGI datasets, CHROM, GREEN, ICA, PBV, POS, SSR, and LGI were experimentally verified. In addition, using the proposed rBS metric, experiments were conducted on 31 areas of the face. The right malar, left malar, glabella, lower medial forehead, and upper medial forehead showed the best results for BVP and BPM extraction. Each area showed a strong correlation with the actual signal, and especially the PCC result was excellent.

Lastly, as the information that can be obtained in one area of the proposed ROI is limited, experiments were conducted on the TOP-5, the entire face, and BOT-5, and the superiority of the TOP-5 was proven. Therefore, it will contribute to effective ROI promotion in the future facial image-based rPPG extraction method, and an improvement of reliability and accuracy of the rPPG method is expected through effective ROI selection.

Existing rPPG methods focused on how well to remove noise from the extracted color information by extracting the color information of the ROIs. Through this study, the superiority of the proposed ROIs using the existing rPPG methods were verified, and it was found that the ROI affects the accuracy of the rPPG method. The rPPG methods that have been conducted so far lack research on the correlation of the ROIs. In a future study, we intend to generate an rPPG algorithm that learns the expression of the correlation in each region using the GNN (Graph Neural Network).

## Figures and Tables

**Figure 1 sensors-21-07923-f001:**
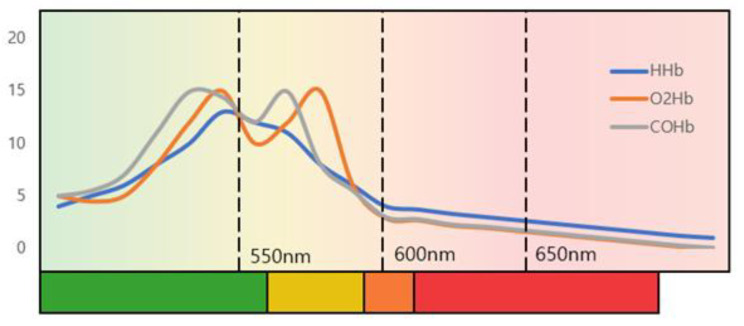
The absorbance of hemoglobin according to the wavelength of light.

**Figure 2 sensors-21-07923-f002:**
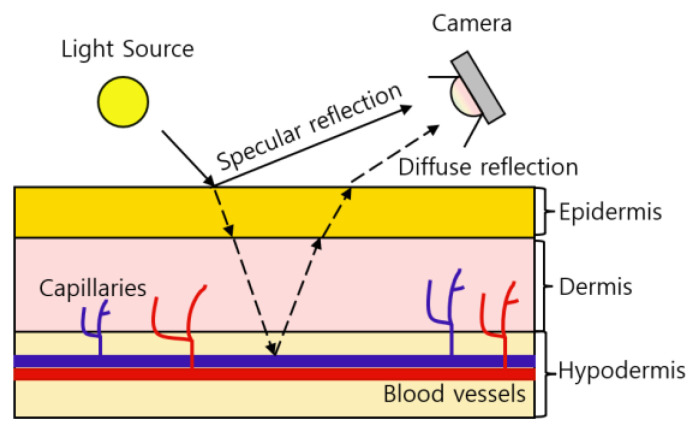
The absorbance of hemoglobin according to the wavelength of light.

**Figure 3 sensors-21-07923-f003:**
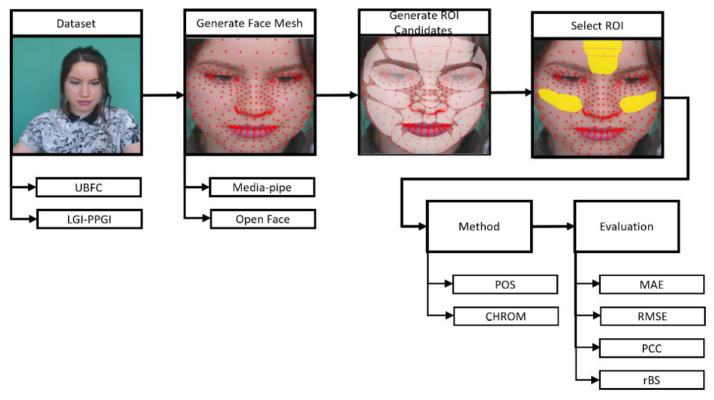
ROI assessment procedure.

**Figure 4 sensors-21-07923-f004:**
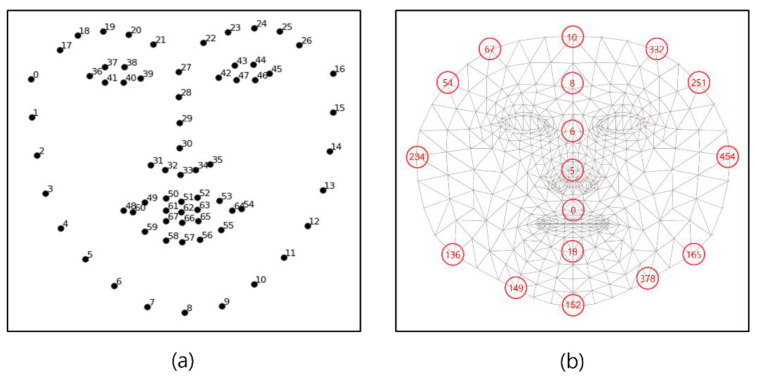
Open-source Face Mesh Projects (**a**) Open Face (**b**) Media-pipe.

**Figure 5 sensors-21-07923-f005:**
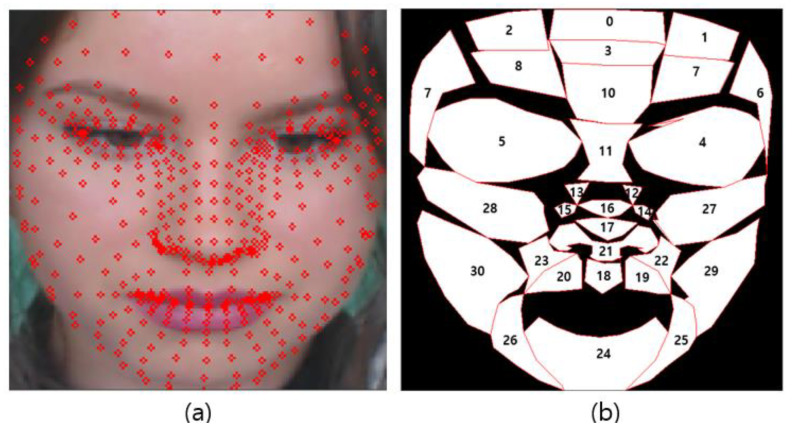
Proposed ROI list (**a**) face mesh image (**b**) ROI index.

**Figure 6 sensors-21-07923-f006:**
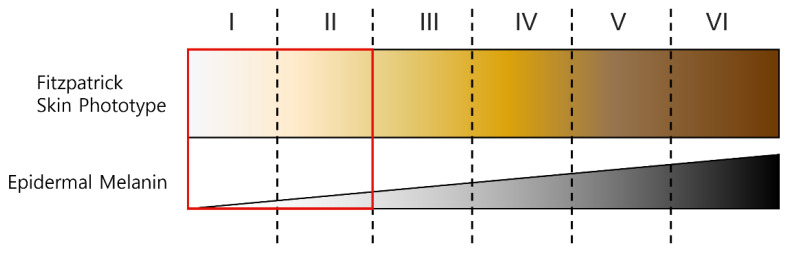
Fitzpatrick skin type and the type used in experiment.

**Figure 7 sensors-21-07923-f007:**
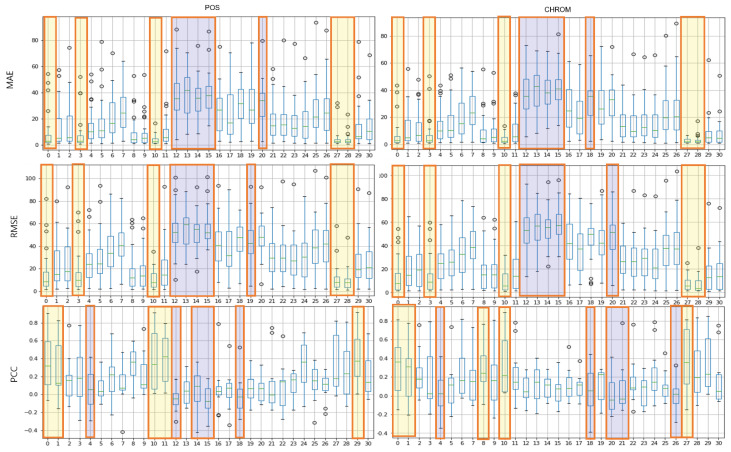
Performance evaluation of proposed ROIs on UBFC data (yellow: TOP-5, blue: BOT-5).

**Figure 8 sensors-21-07923-f008:**
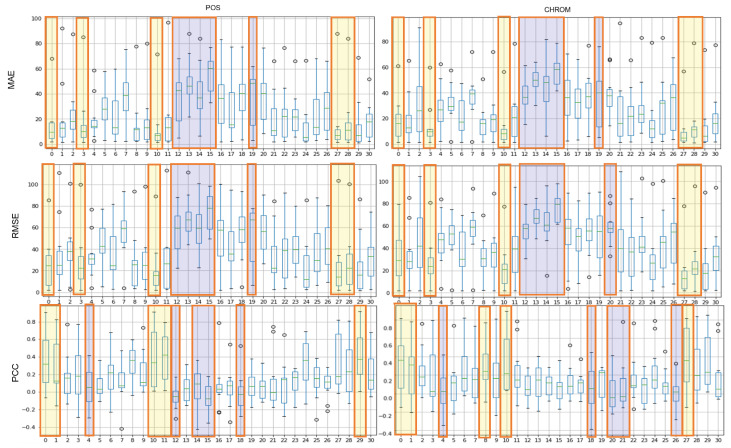
Performance evaluation of proposed ROIs on LGI-PPGI data (yellow: TOP-5, blue: BOT-5).

**Figure 9 sensors-21-07923-f009:**
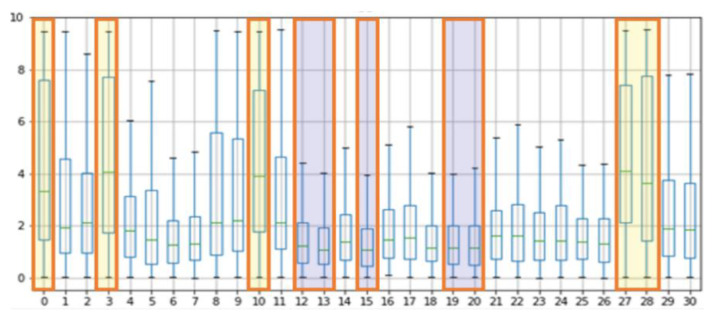
Evaluation of rBS in proposed regions.

**Figure 10 sensors-21-07923-f010:**
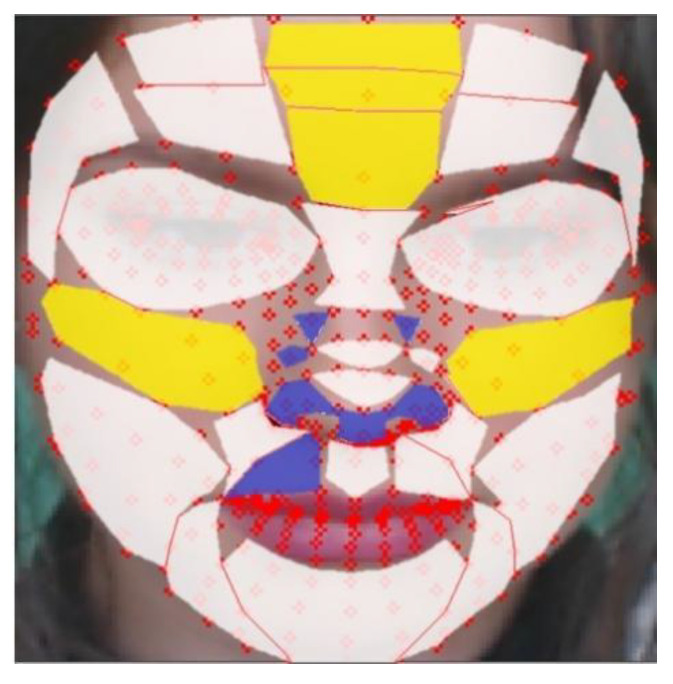
Evaluation of rBS in proposed regions (yellow: TOP-5 regions, blue: BOT-5 regions, white: the other regions).

**Figure 11 sensors-21-07923-f011:**
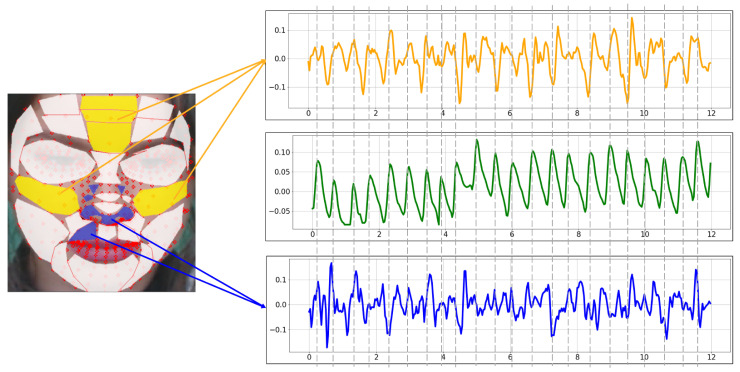
rPPG signal extracted with the proposed ROI using POS (yellow: TOP-5, green: Ground truth, blue: BOT-5).

**Table 1 sensors-21-07923-t001:** Summaries of representative rPPG algorithms.

Method	Characteristic
GREEN [6,7,8,9]	The green channel is preferred for BVP extraction because it has more diffuse reflection information from hemoglobin than other channels. In [17], an attempt was made to visually show the pulse change by maximizing the amount of change in the green channel.
ICA [5]	A method of splitting a multidimensional signal into multiple components. The whitening matrix was obtained using Jacobian rotation, and the actual original signal was separated by multiplying the whitening matrix by the mixed signal. In [5], the mixed signal was separated into four independent components using the JADE method, and empirically, the second signal was used as the PPG signal.
CHROM [10]	The CHROME method removes noise caused by light reflection through color difference channel normalization.
SSR [12]	The SSR method is based on the absorbance of hemoglobin. Using Subspace Rotation and Temporal Rotation has the advantage of extending the pulse amplitude and reducing the distortion by the light reflection.
POS [11]	The POS method aims to reduce the specular noise problem presented by the CHROM to the “plane orthogonal to skin” method. A PPG signal is generated by a projection of the plane orthogonal to skin tone from the temporally normalized RGB plan.
PBV [13]	It suggests a pulse blood vector that distinguishes the pulse-induced color changes from motion noise in the RGB source.
LGI [14]	It suggested a robust algorithm in various environment using differentiable local transformations

**Table 2 sensors-21-07923-t002:** ROI method of each rPPG algorithm.

Method	GREEN	ICA	CHROM	SSR	POS	PBV	LGI
ROI	(1) Face	(1) Face	(1) Face + (2) Skin	(2) Skin	(2) Skin	(2) Skin	(2) Skin

**Table 3 sensors-21-07923-t003:** The dermis and epidermal thickness of 39 facial areas.

Region	Location	Average Epidermal Thickness (μm)	Average Dermal Thickness (μm)	eRT ^(1),^ *	dRT ^(2),^ *	RT ^(3),^ *
0	Upper Medial Forehead	44.70	1200.93	1.51	1.58	1.56
1	Lower Medial Forehead	45.76	1176.11	1.55	1.55	1.53
2	Upper Lateral Forehead	44.80	1252.50	1.52	1.65	1.62
3	Lower Lateral Forehead	39.86	1172.34	1.35	1.54	1.52
4	Upper Medial Eyelid	40.31	758.85	1.36	1.00	1.00
5	Upper Lateral Eyelid	42.39	1088.58	1.43	1.43	1.42
6	Lower Lateral Eyelid	38.58	1227.10	1.30	1.62	1.58
7	Tear Through	47.00	1178.64	1.59	1.55	1.53
8	Glabella	46.59	1339.52	1.58	1.77	1.73
9	Upper Nasal Dorsum	52.19	1475.42	1.77	1.94	1.91
10	Lower Nasal Dorsum	61.60	1198.61	2.08	1.58	1.58
11	Medial Canthus	42.81	840.36	1.45	1.11	1.11
12	Mid Nasal Sidewall	48.45	1746.27	1.64	2.30	2.25
13	Lower Nasal Sidewall	46.70	1969.20	1.58	2.59	2.52
14	ALA	51.57	1941.03	1.74	2.56	2.49
15	Columella	44.17	1160.76	1.49	1.53	1.56
16	Philtrum	48.07	1196.17	1.63	1.58	1.56
17	Nasal Tip	59.77	1288.00	1.68	1.70	1.67
18	Soft Triangle	51.44	1477.47	1.74	1.95	1.91
19	Malar	45.73	1040.46	1.55	1.37	1.36
20	Lower Cheek	44.66	1291.26	1.51	1.70	1.67
21	Upper Lip	62.62	1433.49	2.12	1.89	1.87
22	Nasolabial Fold	48.91	1250.18	1.65	1.65	1.63
23	Marionette Fold	40.87	989.41	1.38	1.30	1.29
24	Chin	45.37	1165.77	1.53	1.54	1.52
25	Temporal	42.18	1245.77	1.43	1.64	1.61
26	Preauricular	37.53	1251.84	1.27	1.65	1.61
27	Upper Helix	42.29	1074.90	1.43	1.42	1.40
28	Mid Helix	56.89	1052.43	1.92	1.39	1.39
29	Conchal Bowl	32.92	999.14	1.11	1.32	1.29
30	Earlobe	44.65	1191.90	1.51	1.57	1.55
31	Lower Medial Eyelid	48.01	868.39	1.62	1.14	1.15
32	Anterior Neck	40.69	1237.68	1.38	1.63	1.60
33	Lateral Neck	32.89	1440.71	1.11	1.90	1.84
34	Posterior Scalp	35.36	1443.86	1.20	2.27	1.85
35	Posterior Auricular	29.57	1724.21	1.00	1.78	2.19
36	Temporal Scalp	33.25	1349.52	1.12	1.51	1.73
37	Anterior Scalp	37.54	1146.13	1.27	1.21	1.48
38	Vertex	37.42	919.45	1.27	1.58	1.20
	Maximum Value	29.57	758.85	2.12	2.58	2.52
	Minimum Value	62.62	1969.20	1.00	1.00	1.00

* are normalized ratios calculated by dividing each thickness by the thinnest value in each category. ^(1)^ the relative thickness of the epidermis. ^(2)^ the relative thickness of the dermis. ^(3)^ the relative thickness.

**Table 4 sensors-21-07923-t004:** Proposed 31 regions.

Region	Location	Thickness (μm)
0	Upper Medial Forehead	1245.63
1	Right Upper Lateral Forehead	1297.30
2	Left Upper Lateral Forehead	1297.30
3	Lower Medial Forehead	1221.88
4	Right Eye	
5	Left Eye	
6	Right Temporal Lobe	1287.96
7	Left Temporal Lobe	1287.96
8	Right Lower Lateral Forehead	1212.20
9	Left Lower Lateral Forehead	1212.20
10	Glabella	1386.11
11	Upper Nasal Dorsum	1527.60
12	Right Mid Nasal Sidewall	1794.71
13	Left Mid Nasal Sidewall	1794.71
14	Right Lower Nasal Sidewall	2015.89
15	Left Lower Nasal Sidewall	2015.89
16	Lower Nasal Dorsum	1496.12
17	Nasal Tip	1496.12
18	Philtrum	1496.12
19	Right Upper Lip	1496.12
20	Left Upper Lip	1496.12
21	Lower Nasal Sidewall	2015.89
22	Right Nasolabial Fold	1299.08
23	Left Nasolabial Fold	1299.08
24	Chin	1211.14
25	Right Marionette Fold	1030.28
26	Left Marionette Fold	1030.28
27	Right Malar	1086.20
28	Left Malar	1086.20
29	Right Lower Cheek	1335.91
30	Left Lower Cheek	1335.91

**Table 5 sensors-21-07923-t005:** Disease information according to frequency band.

Parameter	Frequency	Description
ULF	≤0.003 Hz	Associated with acute heart attack and arrhythmias
VLF	0.033 Hz−0.04 Hz	Variables dependent on the renin–angiotensin system
LF	0.04 Hz–0.15 Hz	Controlled by the sympathetic and parasympathetic nervous systems
HF	0.15 Hz–0.4 Hz	There is a heart rate variability related to the respiratory system, called respiratory arrhythmias

**Table 6 sensors-21-07923-t006:** The median value of rBS and rBS ranking of regions.

Region	0	1	2	3	4	5	6	7	8	9	10
rBS	2.88	1.83	1.81	2.98	1.70	1.32	1.36	1.24	1.97	1.87	3.33
Rank	**4**	9	10	**3**	12	22	19	25	7	8	**2**
**Region**	11	12	13	14	15	16	17	18	19	20	21
rBS	1.98	1.16	1.14	1.33	1.07	1.43	1.36	1.17	1.17	1.16	1.44
Rank	6	28	30	21	31	18	19	26	26	28	17
**Region**	22	23	24	25	26	27	28	29	30		
rBS	1.45	1.49	1.46	1.30	1.27	3.64	2.68	1.81	1.65		
Rank	16	14	15	23	24	**1**	**5**	10	13		

**Table 7 sensors-21-07923-t007:** Thickness and # of pixels at each region.

Region	Location	Thickness μm	# of Pixels	rBS (Rank)
**0**	**Upper Medial Forehead**	**1245.63**	**504**	**4**
1	Right Upper Lateral Forehead	1297.30	389	9
2	Left Upper Lateral Forehead	1297.30	473	10
**3**	**Lower Medial Forehead**	**1221.88**	**454**	**3**
4	Right Eye	-	865	12
5	Left Eye	-	1255	22
6	Right Temporal Lobe	1287.96	17	19
7	Left Temporal Lobe	1287.96	414	25
8	Right Lower Lateral Forehead	1212.20	527	7
9	Left Lower Lateral Forehead	1212.20	597	8
**10**	**Glabella**	**1386.11**	**775**	**2**
11	Upper Nasal Dorsum	1527.60	456	6
12	Right Mid Nasal Sidewall	1794.71	46	28
13	Left Mid Nasal Sidewall	1794.71	57	30
14	Right Lower Nasal Sidewall	2015.89	38	21
15	Left Lower Nasal Sidewall	2015.89	48	31
16	Lower Nasal Dorsum	1496.12	124	18
17	Nasal Tip	1496.12	150	19
18	Philtrum	1496.12	140	26
19	Right Upper Lip	1496.12	179	26
20	Left Upper Lip	1496.12	202	28
21	Lower Nasal Sidewall	2015.89	268	17
22	Right Nasolabial Fold	1299.08	186	16
23	Left Nasolabial Fold	1299.08	213	14
24	Chin	1211.14	990	15
25	Right Marionette Fold	1030.28	312	23
26	Left Marionette Fold	1030.28	408	24
**27**	**Right Malar**	**1086.20**	**794**	**1**
**28**	**Left Malar**	**1086.20**	**955**	**5**
29	Right Lower Cheek	1335.91	840	10
30	Left Lower Cheek	1335.91	1174	13
Correlation coefficient		(Thickness, rBS rank)		0.50
		(# of pixels, rBS rank)		−0.53

**Table 8 sensors-21-07923-t008:** Experimental results for TOP-5, Face + Skin, Bot-5.

	MAE						PCC					
	POS			CHROM			POS			CHROM		
	TOP-5	Face + Skin	BOT-5	TOP-5	Face + Skin	BOT-5	TOP-5	Face + Skin	BOT-5	TOP-5	Face + Skin	BOT-5
UBFC	**1.85**	1.87	7.26	**1.5**	2.67	5.9	0.80	**0.85**	0.29	**0.87**	0.80	0.36
LGI-PPGI	**3.61**	4.04	6.21	**2.93**	4.04	10.71	**0.34**	0.30	0.30	**0.59**	0.38	0.35

## Data Availability

The UBFC dataset is available at https://sites.google.com/view/ybenezeth/ubfcrppg (accessed date 26 November 2021). The LGI-PPGI dataset is available at https://sites.google.com/view/ybenezeth/ubfcrppg (accessed date 26 November 2021).

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
