# Peer review of "Assessment of ROI Selection for Facial Video-Based rPPG"

_sensors, 2021, doi:10.3390/s21237923_

Round 1

Reviewer 1 Report

This paper proposed new ROI for facial video based rPPG and prove how skin thickness affect the accuracy of rPPG algorithm. Authors brought an interesting study and evaluate the performance of their proposed ROI with a great result. However, there are some minor issues that need to be fixed as follows:

  1. Authors should explain more about the background of rPPG assessment. Is regular RGB camera can be used in this study? Is the skin color affect the accuracy of the rPPG assessment?
  2. I think authors can provide the resulting rPPG signal along with the corresponding ground truth in a figure.
  3. There is unnecessary “/” in line 36.
  4. Section 2 only have one subsection, I think authors can just put the content of the subsection in the body of Section 2.
  5. In line 91 authors stated that they propose SSR method, is that correct?
  6. Please provide the detail of LGI method in Section 2.1.7.
  7. There are two Figure 7.
  8. Table 6 is not referred in any paragraph.
  9. Authors should mention the full form of every abbreviation in their first appearance in the manuscript.
  10. Authors should explain the detail of components in every equation like the definition of each variable.

Author Response

Thank you for reviewing my paper.

  1. Authors should explain more about the background of rPPG assessment. Is regular RGB camera can be used in this study? Is the skin color affect the accuracy of the rPPG assessment?

    A: Thanks for the question. In this study, we aim to extract the BVP signal well from a general RGB camera. Restrictions placed on evaluation were not expressed in the thesis, so it was reinforced. The environment restricted up to line 202-218 was specified, and in particular, it was indicated that the melanin content according to skin color affects the results.

  2. I think authors can provide the resulting rPPG signal along with the corresponding ground truth in a figure.

    A : Added rPPG signal result in Figure 12.

  3. There is unnecessary “/” in line 36.

    A : Deleted “/” in line 36.

  4. Section 2 only have one subsection, I think authors can just put the content of the subsection in the body of Section 2.

    A : Recognizing that there are many unnecessary subsections in Section 2, I realized that explaining the sub-items in detail affects the flow of the thesis, so I have summarized them in one table.
  5. In line 91 authors stated that they propose SSR method, is that correct?

    A : I didn't check it properly, and I deleted that part.

  6. Please provide the detail of LGI method in Section 2.1.7.
    A : Since the amount of paper is too large to express the details of the LGI method, it is briefly introduced and replaced with Table 1 in section 2.

  7. There are two Figure 7.

    A : The numbers of all figures were checked and corrected.

  8. Table 6 is not referred in any paragraph.
    A : All tables and figures were checked for referencing, and explanations were added at the bottom of each table and figure.

  9. Authors should mention the full form of every abbreviation in their first appearance in the manuscript.

    A : Added full form for abbreviations such as BVP and BPM.
  10. Authors should explain the detail of components in every equation like the definition of each variable.
    A : As the contents of Section 2 were abbreviated, it is believed that it has been somewhat resolved.

Reviewer 2 Report

The presented paper deals with a study on the effect of skin thickness on the accuracy of the rPPG algorithm, by conducting an experiment using 39 anatomically divided facial regions. The paper is well written and the experiments are well conducted. However, the authors should consider the following points to further improve the quality of the manuscript:

-         The main contributions of the paper should be highlighted in the last part of the introduction (in the form of points).

-         Authors should add some perspectives (future works) in the last part of the conclusion.

-         All parameters in equation (1) should be defined: s, t, E, X. The same remark for the rest of the equations. 

-         Abbreviations should be defined in the first apparition (ex., in the abstract), and some abbreviations are not defined. 

-         Authors should add a graphical flowchart that recapitulates the main modules of the proposed approach.

-         A comparison to some related and recently published papers (under the same conditions: datasets and protocols) should be added.

-        Adding a comparative analysis to some related work in the last remark. 

Author Response

Thanks for reviewing paper.

  1. The main contributions of the paper should be highlighted in the last part of the introduction (in the form of points).
    A : Added major contributions in the form of points (line 62-67,301-305)

  2. Authors should add some perspectives (future works) in the last part of the conclusion.
    A : Future works content has been added to Conclusion. (line322-328)

  3. All parameters in equation (1) should be defined: s, t, E, X. The same remark for the rest of the equations. 
    A : Unnecessary formulas have been removed, and the confusing Section 2 has been simplified. With that, it is presumed that the issue has been resolved.

  4. Abbreviations should be defined in the first apparition (ex., in the abstract), and some abbreviations are not defined. 
    A : The full form of undefined abbreviations such as BVP and BPM has been specified.

  5. Authors should add a graphical flowchart that recapitulates the main modules of the proposed approach.
    A : The main module flowchart of the proposed approach is specified in Figure 3, and the description of the figure is added

  6.  A comparison to some related and recently published papers (under the same conditions: datasets and protocols) should be added.
    A : I searched for papers on the ROI selection method in the rPPG method, but could not find it. I will add more later if I have a chance.

  7. Adding a comparative analysis to some related work in the last remark. 
    A : The final result was visualized in Figure 11 and compared with groundtruth.

Reviewer 3 Report

The article “Assessment of ROI selection for facial video based rPPG” presents a comparison between seven methods to get remote photoplethysmography (rPPG) using two datasets using 31 regions of interest (ROIs) selected after face mesh image in order to associate the results to the skin thickness in each region.

This is an interesting theme for Sensors readers, but the article in general is missing information to become publishable. In order to help in the way to complete the information and correct some typing errors and mistakes, there is a list of points highlighted in the following and in the pdf uploaded.

- Insert a space before the square brackets opening “[” to separate reference citations from the words before it.

- Insert a space before the round brackets opening “(” to separate from the words before it. See for example this “3. ROI(Region Of Interest)”.

- At lines 122 and 123, one literature reference must be cited related to the work mentioned.

- At lines 123 and 124, another literature reference must be cited related to the work mentioned.

- Insert a space after the round brackets closing “)” to separate from the words after it. See for example Table 1 “1)Face” and “2)Skin”

- It is mandatory to define all variables in each equation. It is missing information and compromising the understanding in this way.

- It is strongly recommended to define acronyms in the first time it is cited.

- The text must cite and explain all figures and tables. The correct form is citing figures and tables by the names (for example, Table 2), and not by the position (“and the relative thickness of the skin calculated based on the information are as follows”).

- Turn the notation in Table 3 homogenous putting capital letter in the first letter of each word, using two numbers after decimal point and aligning it by the right.

- The values in Table 2 and Table 3 are missing the unit. Are they measured in mm?

- Correct the equation numbers because there are three equations with the same number (10) at lines 180, 183 and 186.

- The subtitle of Figure 4 is not correct. It is the same as Figure 3 with only (a) and (b) figures but Figure 4 has MAE and RMSE for all 7 methods tested in each of 31 regions using the UFBC data set.

- Figures 4, 5 and 6 should be in better quality. It is not possible to read well in some places.

- It would be great see the highlights of better values found in Figures 4, 5 and 6.  

- At lines 253, 254 e 255 values of thickness must be written using dot not comma (“a skin thickness of (27.1221??)”).

- Figure 7 is not presented in the text. What do yellow, blue and white regions represent? Top five, bottom five and the rest?

- Table 6 is not cited in the text. What is the result presented?

Author Response

Thank you for reviewing my paper.

  1. - Insert a space before the square brackets opening “[” to separate reference citations from the words before it.
    A : Checked all “[” and edited.
  2. - Insert a space before the round brackets opening “(” to separate from the words before it. See for example this “3. ROI(Region Of Interest)”.
    A :  Checked all “(” and edited.
  3. - At lines 122 and 123, one literature reference must be cited related to the work mentioned.
    A : Added [30] reference to that line (line 89).
  4. At lines 123 and 124, another literature reference must be cited related to the work mentioned.
    A : Added [19] reference to that line (line 89-92).
  5. Insert a space after the round brackets closing “)” to separate from the words after it. See for example Table 1 “1)Face” and “2)Skin”
    A : A space is added after every “)” in Table 2.
  6. It is mandatory to define all variables in each equation. It is missing information and compromising the understanding in this way.
    A : Deleted unnecessary formulas in subsections of Section2 and replaced them with brief introductions.
  7. It is strongly recommended to define acronyms in the first time it is cited.
    A : Added full form for abbreviations such as BVP and BPM.
  8. The text must cite and explain all figures and tables. The correct form is citing figures and tables by the names (for example, Table 2), and not by the position (“and the relative thickness of the skin calculated based on the information are as follows”).
    A : According to the points you pointed out, all tables and figures have been modified so that they are referencing.
  9. Turn the notation in Table 3 homogenous putting capital letter in the first letter of each word, using two numbers after decimal point and aligning it by the right.
    A : All words in Table3 have been modified as mentioned.
  10. The values in Table 2 and Table 3 are missing the unit. Are they measured in mm?
    A : The units of Table 2 and Table 3 are indicated. Use values measured in μm
  11. Correct the equation numbers because there are three equations with the same number (10) at lines 180, 183 and 186.
    A : Corrected the number of the formula. (lines 152,153, 156, 159)
  12. The subtitle of Figure 4 is not correct. It is the same as Figure 3 with only (a) and (b) figures but Figure 4 has MAE and RMSE for all 7 methods tested in each of 31 regions using the UFBC data set.
    A : I checked the subtitles and made corrections. (figure 7,8)
  13. Figures 4, 5 and 6 should be in better quality. It is not possible to read well in some places.
    A : It was confirmed that the image was large and could not be seen properly, and the experiment was reduced to make it more visible.
  14. It would be great see the highlights of better values found in Figures 4, 5 and 6.  
    A : The top 5 sets are marked with yellow boxes, and the bottom 5 sets with blue boxes.
  15. At lines 253, 254 e 255 values of thickness must be written using dot not comma (“a skin thickness of (27.1221??)”).
    A : Misunderstanding may occur in that part, so the correction has been modified to show only the thickness. (line 258-262)
  16. Figure 7 is not presented in the text. What do yellow, blue and white regions represent? Top five, bottom five and the rest?
    A : Added description for Figure 7 (now Figure 10) on lines 266-268.
  17. Table 6 is not cited in the text. What is the result presented?
    A : Added explanation for Table 6 (currently Table 8). (line 25-291)

Reviewer 4 Report

Assessment of ROI selection for facial video based rPPG is considered as one of the facial landmark measurement methods, and it is a very interesting study.

In particular, the authors' experiments on 39 anatomically divided facial regions for the effect of skin thickness on the accuracy of the rPPG algorithm are considered to be of academic contribution.

- Comparing the effectiveness of the 31 suggested areas with some of the following referenced landmarks is expected to be a better study.

-Face detection and attributes, URL: https://docs.microsoft.com/en-us/azure/cognitive-services/face/concepts/face-detection
- Eason, Face Landmarks Analysis with Azure Face Service and OpenCV, URL: https://medium.com/analytics-vidhya/face-landmarks-analysis-with-azure-face-service-and-opencv-fb898ab58929
- D. Zhou, D. Petrovska-Delacrétaz and B. Dorizzi, "Automatic landmark location with a Combined Active Shape Model," 2009 IEEE 3rd International Conference on Biometrics: Theory, Applications, and Systems, 2009, pp. 1-7, doi: 10.1109/BTAS.2009.5339037.
- Hung Phuoc Truong, Quan Manh Le, Thinh Long Nguyen, Yong-Guk Kim, “Facial landmarks detection for evaluating facial paralysis using a modern active shape model“, 2018
- Adrian Rosebrock, Facial landmarks with dlib, OpenCV, and Python, 2017. URL: https://www.pyimagesearch.com/2017/04/03/facial-landmarks-dlib-opencv-python/

minor points
- The image in Figure 4,5,6 is blurry. Please present a clear image.

Author Response

Thank you for reviewing my paper.

  1. Comparing the effectiveness of the 31 suggested areas with some of the following referenced landmarks is expected to be a better study.
    A : As suggested, a brief mention of the face landmark was added. (line 183-198)
  2. The image in Figure 4,5,6 is blurry. Please present a clear image.
    A : Removed some experiments from the picture and increased visibility.(Figure 7,8,9)

Reviewer 5 Report

In this paper, the authors investigated the relationship between skin thickness and accuracy of facial image-based rPPG methods. Therefore, the face is divided into 31 facial regions based on 39 anatomically classified regions and experiments are performed considering 7 algorithms, a metric proposed by the authors is used to identify the most accurate regions for BVP and BPM extraction. This article deals with an interesting topic but, in my opinion, it needs to be greatly improved. Currently, it looks more like a draft that needs to be expanded and corrected. Indeed, even if the general organization is decent, the content is quite problematic for several reasons.

First, some parts are too vague, in particular the description of the 7 algorithms in which many notations are unexplained and not uniformized. For example, what are X and E in Equation (1)? What is the range for alpha in Equation (3), or sigma and mu in Equation (5) and (6), and so on. Same remark for N in Equations (9) and (10). In Equation (7) you introduce X_r, X_g and X_b, are they different from R_s, G_s and B_s used previously?  Moreover, still in section 2, some sentences describing the algorithms are really quite unclear or need to be completed:

* page 2 - lines 74-75 => 'Construct... as a time series'

* page 2 - lines 80-81 => Why did the authors of [5] use the second signal?

* page 3 - line 91 => 'We propose "Spatial Subspace Rotation"...' Do 'We' mean you?

* page 3 - line 111 => You need to say more about 'a model space for the generalization ability of the learning algorithm is proposed'.

In summary, the different algorithms are described too briefly and a discussion of the advantages/disadvantages of each, or a comparison, could conclude section 2. The contributions should also appear more clearly in the introduction - Section 1.

Second, Figures 4, 5 and 6 are quite unclear and the comments in Section 4.2 quite questionable. Why are regions 10 and 28 not noticed in flax 222? Also, the legends in Figures 4 and 7 are problematic. The second in the latter figure, because there are two figures 7 (the first line 244 and the second line 286). Also, there is absolutely no discussion of Figure 7 (the second one with the face at the bottom of page 12) and Table 6. Similarly, Figure 1 is no cited and there are two figures 1(the first line 53 and the second line 143). What do the two colors in the figure line 286 mean? Which color designates the Top-5 regions (same remark for Bot-5)? You should also systematically add the units of the values shown in the various tables when it makes sense.

Third, the paragraph analyzing the values shown in Table 5 (page 12 - lines 251-255), and in particular the link between scoring and skin thickness is not obvious. Indeed, the two regions with the highest scores have the greatest thickness. The values given for thicknesses in this paragraph are also very different from those noted in Table 3. In fact, the order of magnitude is definitely not the same.

Four, there are many spelling mistakes, misused notations, problems due to copy/paste that make reading difficult. There are also some English language problems that need to be corrected, with some sentences that are not clear at all as noted above. Many acronyms are used/introduced among which many are not clearly defined.

To conclude, I am not completely convinced by the results and in my opinion the document is poorly written. The document needs to be really greatly improved and therefore I recommend a major revision.

I noticed below some various problems with sometimes some suggestion (this is not an exhaustive list):

* page 1 – line 36 => '98\%' should be 98%

* page 1 – line 42 => 'fPPG, and it can…' could be replaced by 'fPPG, it can…'

* page 2 – line 57 => 'for designating a designated area' -> 'for designating a chosen area'

* page 2 – lines 63-65 => Is it 'Section' and not 'Session' and I think that 'method' and 'algorithm' should be plural.

* page 2 – line 70 => Revise subsection numbering in Section 2. I think that 2.1 is a useless numbering level.

You should thus change 2.1.1 in 2.1, 2.1.2 in 2.2 and so on.

* page 4 – line 126 => I suggest to add the following sentence at the end 'Table 1 sums-up the ROI method used by each rPPG algorithm.', 'proposed area' -> 'proposed areas'

* page 4 – Caption of Table 1 => 'rPPG algorithms' -> 'rPPG algorithm'

* page 4 – line 143 - Figure 1 => Should be Figure 2

* etc.

Author Response

Thanks for reviewing my paper.

After reading the review, I took a closer look, and realized that there was a mistake in the experiment, and I am grateful for it.

  • First
    A : The content of Section 2 is too broad to be included in the thesis, and it was judged that it was out of topic to express all the formulas.
    I added details about the experimental environment in Section 3, and added the main contribution in the form of points at the end of Section 1.

  • Second
    A : After seeing the contents of Second, we conducted a code review, found errors and corrected them. The numbers of all figures and tables have been corrected, and explanations have been added by referencing for better understanding. Also, in the case of Figure 10, which was a problem, it was actually wrongly marked, and it was corrected according to the experimental results. Finally, for easy understanding of all figures, yellow indicates high accuracy and blue indicates low accuracy.
  • Third
    As pointed out in Second, the experiment was re-run, recognizing that the relationship with the skin thickness was not clear. As a result, there was an error in the experimental process, and the correlation coefficient was added at the bottom of Table 7 to evaluate the effect.

  • Fourth
    Non-uniform expressions such as roi, Roi, and ROI were unified into one. In addition, abbreviations that were not written in full form such as BVP, BPM, and fPPG were specified in full form. Finally, the part specified as ROI region was changed to ROI, and the part mixed with BVP and rPPG signal was unified into BVP.

Round 2

Reviewer 2 Report

I am OK with the response from the authors. The manuscript can be considered for an acceptance.

Author Response

Thanks for the second review.

I fixed some English miss typing and Table form.

video based -> video-based (line2)
remote PPG(rPPG) -> remote PPG (rPPG) (line42)
face PPG(rPPG) -> face PPG (rPPG) (line42)
absorbance -> The absorbance (line 54)
ROI of the Section 2's algorithms  -> ROI of Section 2's algorithms (line 71)
The SSR method based on absorbance of hemoglobin -> The SSR method is based on the absorbance of hemoglobin (Table1)
It suggests Pulse Blood Vector that distinguish the pulse-induced col-or changes from motion noise in RGB source.
-> It suggests Pulse Blood Vector that distinguishes the pulse-induced col-or changes from motion noise in RGB source.
It suggested the robust algorithm in various environment using differentiable local transformations -> It suggested the robust algorithm in the various environment using differentiable local transformations
are reflections due to absorption and scattering of skin tissue that -> are reflections due to the absorption and scattering of skin tissue that 
absorbance of hemoglobin according to the wavelength of light.
-> The absorbance of hemoglobin according to the wavelength of light. (figure 2)
The face mesh extraction method is largely divided into a cascaded regression based and deep learning based methods.
-> The face mesh extraction method is largely divided into cascaded regression based and deep learning-based methods.(line 185)
Figure 4 is a face landmark key point of cascaded regression based Open Face Project and deep learning-based media-pipe project ->
Figure 4 is a face landmark key point of cascaded regression-based Open Face Project and deep learning-based media-pipe project(line 189)
A proposed ROI mask was generated fortwo datasets,
-> A proposed ROI mask was generated for two datasets,
number -> numbers(line 233,234)
Figure 7 shows the results of MAE, RMSE, and PCC metric on UBFC data. 
-> Figure 7 shows the results of MAE, RMSE, and PCC metrics on UBFC data. (luine 240)
Figure 8 shows the results of MAE, RMSE, and PCC metric on LGI-PPGI data.
-> Figure 8 shows the results of MAE, RMSE, and PCC metrics on LGI-PPGI data.(line 248)
and the number of pixel's -> and the number of pixels (line 279)

Reviewer 5 Report

The paper has been improved in a suitable way.  The authors have addressed the different points raised in the review convincingly and particularly the ones dealing with technical aspects. More details are also given on some key points. There are still some minor corrections to be done as noticed in the attached file. However, I am satisfied with this revision and I recommend acceptance of the paper after minor revision.

Author Response

Thanks for the second review.

The paper was extensively revised by referring to the attached file.

And I checked and corrected some grammar points.

Thanks for the English style guide.

video based -> video-based (line2)
remote PPG(rPPG) -> remote PPG (rPPG) (line42)
face PPG(rPPG) -> face PPG (rPPG) (line42)
absorbance -> The absorbance (line 54)
ROI of the Section 2's algorithms  -> ROI of Section 2's algorithms (line 71)
The SSR method based on absorbance of hemoglobin -> The SSR method is based on the absorbance of hemoglobin (Table1)
It suggests Pulse Blood Vector that distinguish the pulse-induced col-or changes from motion noise in RGB source.
-> It suggests Pulse Blood Vector that distinguishes the pulse-induced col-or changes from motion noise in RGB source.
It suggested the robust algorithm in various environment using differentiable local transformations -> It suggested the robust algorithm in the various environment using differentiable local transformations
are reflections due to absorption and scattering of skin tissue that -> are reflections due to the absorption and scattering of skin tissue that 
absorbance of hemoglobin according to the wavelength of light.
-> The absorbance of hemoglobin according to the wavelength of light. (figure 2)
The face mesh extraction method is largely divided into a cascaded regression based and deep learning based methods.
-> The face mesh extraction method is largely divided into cascaded regression based and deep learning-based methods.(line 185)
Figure 4 is a face landmark key point of cascaded regression based Open Face Project and deep learning-based media-pipe project ->
Figure 4 is a face landmark key point of cascaded regression-based Open Face Project and deep learning-based media-pipe project(line 189)
A proposed ROI mask was generated fortwo datasets,
-> A proposed ROI mask was generated for two datasets,
number -> numbers(line 233,234)
Figure 7 shows the results of MAE, RMSE, and PCC metric on UBFC data. 
-> Figure 7 shows the results of MAE, RMSE, and PCC metrics on UBFC data. (luine 240)
Figure 8 shows the results of MAE, RMSE, and PCC metric on LGI-PPGI data.
-> Figure 8 shows the results of MAE, RMSE, and PCC metrics on LGI-PPGI data.(line 248)
and the number of pixel's -> and the number of pixels (line 279)